# QWEN-VL: A VERSATILE VISION-LANGUAGE MODEL FOR UNDERSTANDING, LOCALIZATION, TEXT READING, AND BEYOND

## ABSTRACT

In this work, we introduce the Qwen-VL series, a set of large-scale vision-language models (LVLMs) designed to perceive and understand both texts and images. Starting from the Qwen-LM as a foundation, we endow it with visual capacity by the meticulously designed (i) visual receptor, (ii) input-output interface, (iii) 3-stage training pipeline, and (iv) multilingual multimodal cleaned corpus. Beyond the conventional image description and question-answering, we implement the grounding and text-reading ability of Qwen-VLs by aligning image-caption-box tuples. The resulting models, including Qwen-VL and Qwen-VL-Chat, set new records for generalist models under similar model scales on a broad range of visual-centric benchmarks (*e.g.*, image captioning, question answering, visual grounding) and different settings (*e.g.*, zero-shot, few-shot). Moreover, on real-world dialog benchmarks, our instruction-tuned Qwen-VL-Chat also demonstrates superiority compared to existing vision-language chatbots. All models are public to facilitate future research.

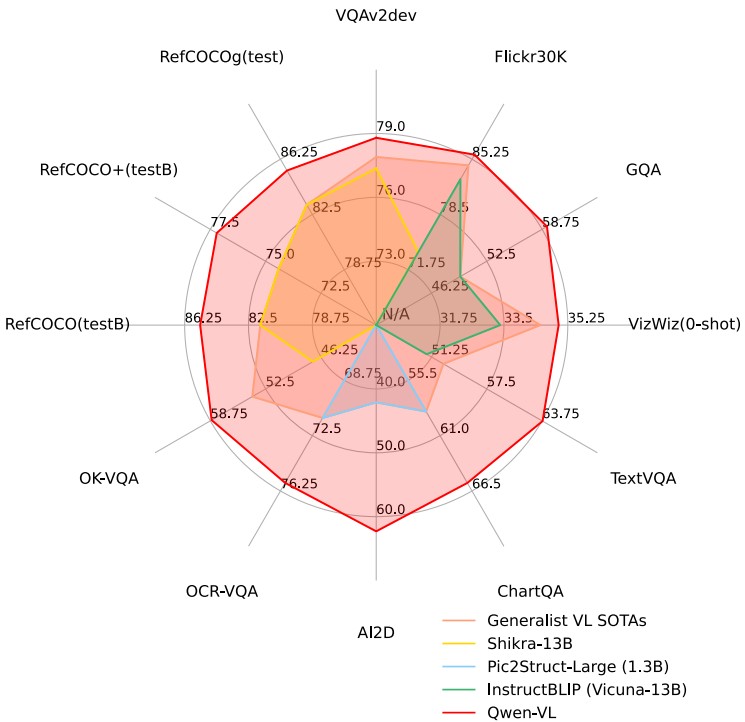

Figure 1: Qwen-VL achieves state-of-the-art performance on a broad range of tasks compared with other generalist models.

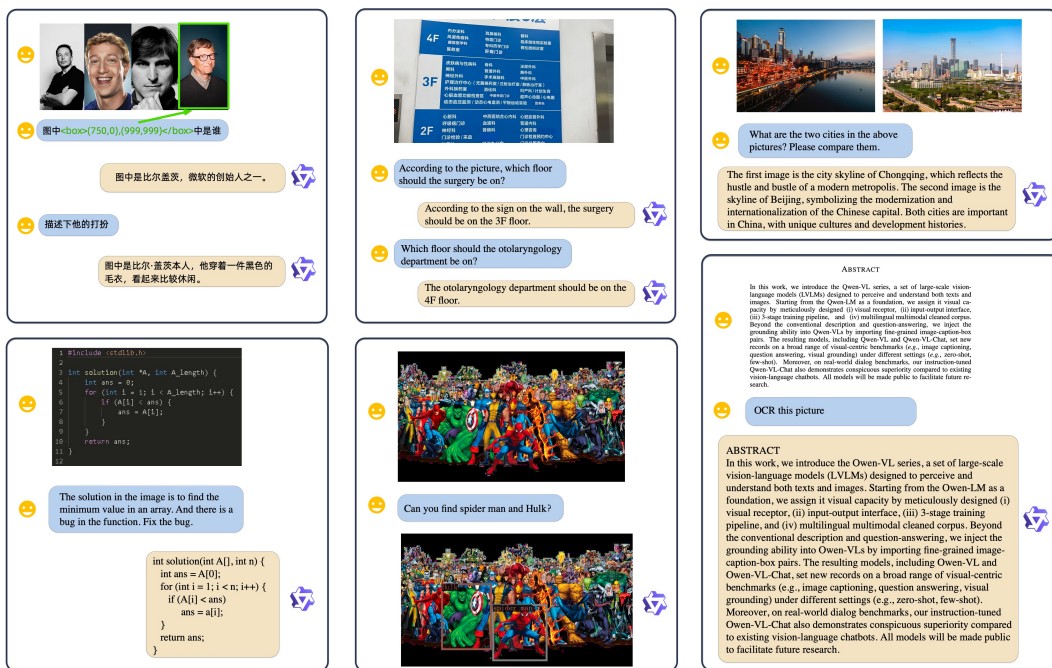

Figure 2: Some qualitative examples generated by our Qwen-VL-Chat. Qwen-VL-Chat supports multiple image inputs, multi-round dialogue, multilingual conversation, text-reading, localization, fine-grained recognition and understanding ability.

# 1 INTRODUCTION

Recently, Large Language Models (LLMs) (Brown et al., 2020; OpenAI, 2023; Anil et al., 2023; Gao et al., 2023; Qwen, 2023) have attracted wide attention due to their powerful capabilities in text generation and comprehension. These models can be further aligned with user intent through fine-tuning instructions, showcasing strong interactive capabilities and the potential to enhance productivity as intelligent assistants. However, native large language models only live in the pure-text world, lacking the ability to handle other common modalities (such as images, speech, and videos), resulting in great restrictions on their application scope. Motivated by this, a group of Large Vision Language Models (LVLMs) (Alayrac et al., 2022; Chen et al., 2022; Li et al., 2023c; Dai et al., 2023; Huang et al., 2023; Peng et al., 2023; Zhu et al., 2023; Liu et al., 2023; Ye et al., 2023b;a; Chen et al., 2023a; Li et al., 2023a; Zhang et al., 2023; Sun et al., 2023; OpenAI, 2023) have been developed to enhance large language models with the ability to perceive and understand visual signals. These large-scale vision-language models demonstrate promising potential in solving real-world vision-central problems.

Nevertheless, despite that lots of works have been conducted to explore the limitation and potency of LVLMs, current open-source LVLMs always suffer from inadequate training and optimization, thus lag far behind the proprietary models (Chen et al., 2022; 2023b; OpenAI, 2023), which hinders further exploration and application of LVLMs in open-source community. What's more, as real-world visual scenarios are quite complicated, fine-grained visual understanding plays a crucial role for LVLMs to assist people effectively and precisely. But only a few attempts had been made toward this direction (Peng et al., 2023; Chen et al., 2023a), the majority of open-source LVLMs remain perceiving the image in a coarse-grained approach and lacking the ability to execute fine-grained perception such as object grounding or text reading.

In this paper, we explore a way out and present the newest members of the open-sourced Qwen families: Qwen-VL series. Qwen-VLs are a series of highly performant and versatile vision-language foundation models based on Qwen-7B (Qwen, 2023) language model. We empower the LLM basement with visual capacity by introducing a new visual receptor including a language-aligned visual encoder and a position-aware adapter. The overall model architecture as well as the input-output

interface are quite concise and we elaboratedly design a 3-stage training pipeline to optimize the whole model upon a vast collection of image-text corpus. Our pre-trained checkpoint, termed Qwen-VL, is capable of perceiving and understanding visual inputs, generating desired responses according to given prompts, and accomplishing various vision-language tasks such as image captioning, question answering, text-oriented question answering, and visual grounding. Qwen-VL-Chat is the instruction-tuned vision-language chatbot based on Qwen-VL. As shown in Fig. 2, Qwen-VL-Chat is able to interact with users and perceive the input images following the intention of users. Specifically, the features of the Qwen-VL series models include:

- Leading performance: Qwen-VLs achieve top-tier accuracy on a vast of vision-centric understanding benchmarks compared to counterparts with similar scales. Besides, Qwen-VL's stuning performance covers not only the conventional benchmarks *e.g.*, captioning, question-answering, grounding), but also some recently introduced dialogue benchmarks.

- Multi-lingual: Similar to Qwen-LM, Qwen-VLs are trained upon multilingual image-text data with a considerable amount of corpus being in English and Chinese. In this way, Qwen-VLs naturally support English, Chinese, and multilingual instructions.

- Multi-image: In the training phase, we allow arbitrary interleaved image-text data as Qwen-VL's inputs. This feature allows our Qwen-Chat-VL to compare, understand, and analyze the context when multiple images are given.

- Fine-grained visual understanding: Thanks to the higher-resolution input size and fine-grained corpus we used in training, Qwen-VLs exhibit highly competitive fine-grained visual understanding ability. Compared to existing vision-language generalists, our Qwen-VLs possess much better grounding, text-reading, text-oriented question answering, and fine-grained dialog performance.

## 2 METHODOLOGY

### 2.1 MODEL ARCHITECTURE

The overall network architecture of Qwen-VL consists of three components and the details of model parameters are shown in Table 1:

**Large Language Model**: Qwen-VL adopts a large language model as its foundation component. The model is initialized with pre-trained weights from Qwen-7B (Qwen, 2023).

**Visual Encoder**: The visual encoder of Qwen-VL uses the Vision Transformer (ViT) (Dosovitskiy et al., 2021) architecture, initialized with pre-trained weights from Openclip's ViT-bigG (Ilharco et al., 2021). During both training and inference, input images are resized to a specific resolution. The visual encoder processes images by splitting them into patches with a stride of 14, generating a set of image features.

**Position-aware Vision-Language Adapter**: To alleviate the efficiency issues arising from long image feature sequences, Qwen-VL introduces a vision-language adapter that compresses the image features. This adapter comprises a single-layer cross-attention module initialized randomly. The module uses a group of trainable vectors (Embeddings) as query vectors and the image features from the visual encoder as keys for cross-attention operations. This mechanism compresses the visual feature sequence to a fixed length of 256. The ablation about the number of queries is shown in Appendix E.2. Additionally, considering the significance of positional information for fine-grained image comprehension, 2D absolute positional encodings are incorporated into the cross-attention mechanism's query-key pairs to mitigate the potential loss of positional details during compression. The compressed image feature sequence of length 256 is subsequently fed into the large language model.

Table 1: Details of Qwen-VL model parameters.

| Vision Encoder | VL Adapter | LLM | Total |
|---|---|---|---|
| 1.9B | 0.08B | 7.7B | 9.6B |

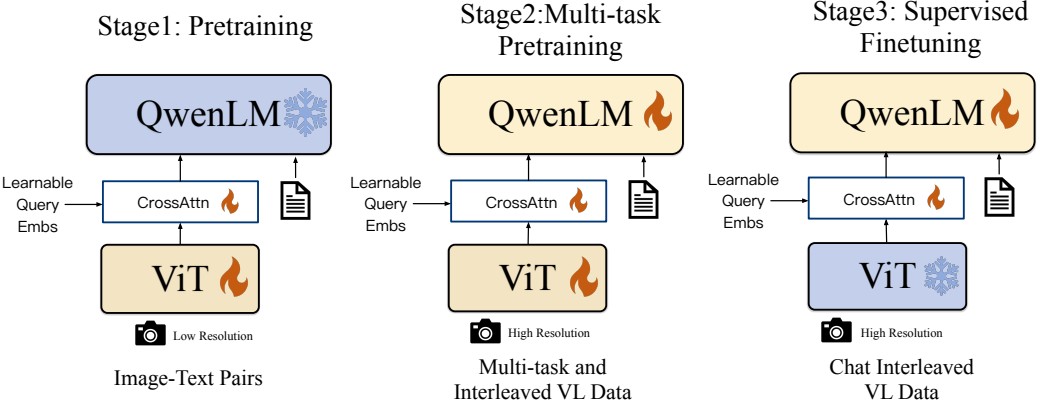

Figure 3: The training pipeline of the Qwen-VL series.

## 2.2 INPUTS AND OUTPUTS

**Image Input**: Images are processed through the visual encoder and adapter, yielding fixed-length sequences of image features. To differentiate between image feature input and text feature input, two special tokens ( and </img>) are appended to the beginning and end of the image feature sequence respectively, signifying the start and end of image content.

**Bounding Box Input and Output**: To enhance the model's capacity for fine-grained visual understanding and grounding, Qwen-VL's training involves data in the form of region descriptions, questions, and detections. Differing from conventional tasks involving image-text descriptions or questions, this task necessitates the model's accurate understanding and generation of region descriptions in a designated format. For any given bounding box, a normalization process is applied (within the range [0, 1000)) and transformed into a specified string format: "$(X_{topleft}, Y_{topleft}), (X_{bottomright}, Y_{bottomright})$". The string is tokenized as text and does not require an additional positional vocabulary. To distinguish between detection strings and regular text strings, two special tokens (<box> and </box> are added at the beginning and end of the bounding box string. Additionally, to appropriately associate bounding boxes with their corresponding descriptive words or sentences, another set of special tokens (<ref> and </ref>) is introduced, marking the content referred to by the bounding box.

## 3 TRAINING

As illustrated in Fig. 3, the training process of the Qwen-VL model consists of three stages: two stages of pre-training and a final stage of instruction fine-tuning training.

## 3.1 PRE-TRAINING

In the first stage of pre-training, we mainly utilize a large-scale, weakly labeled, web-crawled set of image-text pairs. Our pre-training dataset is composed of several publicly accessible sources and some in-house data. We made an effort to clean the dataset of certain patterns. As summarized in Table 2, the original dataset contains a total of 5 billion image-text pairs, and after cleaning, 1.4 billion data remain, with 77.3% English (text) data and 22.7% Chinese (text) data.

We freeze the large language model and only optimize the vision encoder and VL adapter in this stage. The input images are resized to $224 \times 224$. The training objective is to minimize the cross-entropy of the text tokens. The maximum learning rate is $2e^{-4}$ and the training process uses a batch size of 30720 for the image-text pairs, and the entire first stage of pre-training lasts for 50,000 steps, consuming approximately 1.5 billion image-text samples. More hyperparameters are detailed in Appendix C and the convergence curve of this stage is shown in Figure 6.

Table 2: Details of Qwen-VL pre-training data. LAION-en and LAION-zh are the English and Chinese language subset of LAION-5B (Schuhmann et al., 2022a). LAION-COCO (Schuhmann et al., 2022b) is a synthetic dataset generated from LAION-en. DataComp (Gadre et al., 2023) and Coyo (Byeon et al., 2022) are collections of image-text pairs. CC12M (Changpinyo et al., 2021), CC3M (Sharma et al., 2018), SBU (Ordonez et al., 2011) and COCO Caption (Chen et al., 2015) are academic caption datasets.

| Language | Dataset | Original | Cleaned | Remaining% |
|----------|---------|----------|---------|------------|
| English | LAION-en | 2B | 280M | 14% |
| | LAION-COCO | 600M | 300M | 50% |
| | DataComp | 1.4B | 300M | 21% |
| | Coyo | 700M | 200M | 28% |
| | CC12M | 12M | 8M | 66% |
| | CC3M | 3M | 3M | 100% |
| | SBU | 1M | 0.8M | 80% |
| | COCO Caption | 0.6M | 0.6M | 100% |
| Chinese | LAION-zh | 108M | 105M | 97% |
| | In-house Data | 220M | 220M | 100% |
| | Total | 5B | 1.4B | 28% |

## 3.2 MULTI-TASK PRE-TRAINING

In the second stage of multi-task pre-training, we introduce high-quality and fine-grained VL annotation data with a larger input resolution and interleaved image-text data. As summarized in Table 3, we trained Qwen-VL on 7 tasks simultaneously. For text generation, we use the in-house collected corpus to maintain the LLM's ability. Captioning data is the same with Table 2 except for far fewer samples and excluding LAION-COCO. We use a mixture of publicly available data for the VQA task which includes GQA (Hudson & Manning, 2019), VGQA (Krishna et al., 2017), VQAv2 (Goyal et al., 2017), DVQA (Kafle et al., 2018), OCR-VQA (Mishra et al., 2019) and DocVQA (Mathew et al., 2021). We follow Kosmos-2 to use the GRIT (Peng et al., 2023) dataset for the grounding task with minor modifications. For the reference grounding and grounded captioning duality tasks, we construct training samples from GRIT (Peng et al., 2023), Visual Genome (Krishna et al., 2017), RefCOCO (Kazemzadeh et al., 2014), RefCOCO+, and RefCOCOg (Mao et al., 2016). In order to improve the text-oriented tasks, we collect pdf and HTML format data from Common Crawl[1] and generate synthetic OCR data in English and Chinese language with natural scenery background, following (Kim et al., 2022). Finally, we simply construct interleaved image-text data by packing the same task data into sequences of length 2048.

Table 3: Details of Qwen-VL multi-task pre-training data.

| Task | # Samples | Dataset |
|------|-----------|---------|
| Captioning | 19.7M | LAION-en & zh, DataComp, Coyo, CC12M & 3M, SBU, COCO, In-house Data |
| VQA | 3.6M | GQA, VGQA, VQAv2, DVQA, OCR-VQA, DocVQA, TextVQA, ChartQA, AI2D |
| Grounding[2] | 3.5M | GRIT |
| Ref Grounding | 8.7M | GRIT, Visual Genome, RefCOCO, RefCOCO+, RefCOCOg |
| Grounded Cap. | 8.7M | GRIT, Visual Genome, RefCOCO, RefCOCO+, RefCOCOg |
| OCR | 24.8M | SynthDoG-en & zh, Common Crawl pdf & HTML |
| Pure-text Autoregression | 7.8M | In-house Data |

We increase the input resolution of the visual encoder from $224 \times 224$ to $448 \times 448$, reducing the information loss caused by image down-sampling. Besides, we ablate the window attention and

---

[1] https://digitalcorpora.org/corpora/file-corpora/cc-main-2021-31-pdf-untruncated

[2] This task is to generate noun/phrase grounded captions (Peng et al., 2023).

Table 4: Results on Image Captioning and General VQA.

| Model Type | Model | Image Caption | | General VQA | | | | |
|---|---|---|---|---|---|---|---|---|
| | | Nocaps (0-shot) | Flickr30K (0-shot) | VQAv2 | OKVQA | GQA | SciQA-Img (0-shot) | VizWiz (0-shot) |
| Generalist Models | Flamingo-9B | - | 61.5 | 51.8 | 44.7 | - | - | 28.8 |
| | Flamingo-80B | - | 67.2 | 56.3 | 50.6 | - | - | 31.6 |
| | Unified-IO-XL | 100.0 | - | 77.9 | 54.0 | - | - | - |
| | Kosmos-1 | - | 67.1 | 51.0 | - | - | - | 29.2 |
| | Kosmos-2 | - | 80.5 | 51.1 | - | - | - | - |
| | BLIP-2 (Vicuna-13B) | 103.9 | 71.6 | 65.0 | 45.9 | 32.3 | 61.0 | 19.6 |
| | InstructBLIP (Vicuna-13B) | **121.9** | 82.8 | - | - | 49.5 | 63.1 | 33.4 |
| | Shikra (Vicuna-13B) | - | 73.9 | 77.36 | 47.16 | - | - | - |
| | **Qwen-VL (Qwen-7B)** | 121.4 | **85.8** | **79.5** | **58.6** | **59.3** | 67.1 | 35.2 |
| | **Qwen-VL-Chat** | 120.2 | 81.0 | 78.2 | 56.6 | 57.5 | **68.2** | **38.9** |
| Specialist SOTAs | - | 127.0 (PALI-17B) | 84.5 (InstructBLIP -FlanT5-XL) | 86.1 (PALI-X -55B) | 66.1 (PALI-X -55B) | 72.1 (CFR) | 92.53 (LLaVa+ GPT-4) | 70.9 (PALI-X -55B) |

global attention for higher resolutions of the vision transformer in Appendix E.3. We unlocked the large language model and trained the whole model. The training objective is the same as the pre-training stage.

### 3.3 SUPERVISED FINE-TUNING

During this stage, we finetuned the Qwen-VL pre-trained model through instruction fine-tuning to enhance its instruction following and dialogue capabilities, resulting in the interactive Qwen-VL-Chat model. The multi-modal instruction tuning data primarily comes from caption data or dialogue data generated through LLM self-instruction, which often only addresses single-image dialogue and reasoning and is limited to image content comprehension. We construct an additional set of dialogue data through manual annotation, model generation, and strategy concatenation to incorporate localization and multi-image comprehension abilities into the Qwen-VL model. We confirm that the model effectively transfers these capabilities to a wider range of languages and question types. Additionally, we mix multi-modal and pure text dialogue data during training to ensure the model's universality in dialogue capabilities. The instruction tuning data amounts to 350k. In this stage, we freeze the visual encoder and optimize the language model and adapter module. We demonstrate the data format of this stage in Appendix B.2.

## 4 EVALUATION

In this section, we conduct an overall evaluation on various multi-modal tasks to comprehensively assess our models' visual understanding ability. In the following, Qwen-VL denotes the model after the multi-task training, and Qwen-VL-Chat denotes the model after supervised fine-tuning (SFT) stage. Table 9 provides a detailed summary of the used evaluation benchmarks and corresponding metrics.

### 4.1 IMAGE CAPTION AND GENERAL VISUAL QUESTION ANSWERING

Image caption and general visual question answering (VQA) are two conventional tasks for vision-language models. Specifically, image caption requires the model to generate a description for a given image and general VQA requires the model to generate an answer for a given image-question pair.

For the image caption task, we choose Nocaps (Agrawal et al., 2019) and Flickr30K (Young et al., 2014) as benchmarks and report CIDEr score (Vedantam et al., 2015) as metric. We utilize greedy search for caption generation with a prompt of *"Describe the image in English:"*.

For general VQA, we utilize five benchmarks including VQAv2 (Goyal et al., 2017), OKVQA (Marino et al., 2019), GQA (Hudson & Manning, 2019), ScienceQA (Image Set) (Lu et al., 2022b) and VizWiz VQA (Gurari et al., 2018). For VQAv2, OKVQA, GQA and VizWiz

Table 5: Results on Text-oriented VQA.

| Model type | Model | TextVQA | DocVQA | ChartQA | AI2D | OCR-VQA |
|---|---|---|---|---|---|---|
| Generalist Models | BLIP-2 (Vicuna-13B) | 42.4 | - | - | - | - |
| | InstructBLIP (Vicuna-13B) | 50.7 | - | - | - | - |
| | mPLUG-DocOwl (LLaMA-7B) | 52.6 | 62.2 | 57.4 | - | - |
| | Pix2Struct-Large (1.3B) | - | **76.6** | 58.6 | 42.1 | 71.3 |
| | **Qwen-VL (Qwen-7B)** | **63.8** | 65.1 | 65.7 | **62.3** | **75.7** |
| | **Qwen-VL-Chat** | 61.5 | 62.6 | **66.3** | 57.7 | 70.5 |
| Specialist SOTAs | PALI-X-55B (Single-task fine-tuning, without OCR Pipeline) | 71.44 | 80.0 | 70.0 | 81.2 | 75.0 |

VQA, we employ open-ended answer generation with greedy decoding strategy and a prompt of *"{question} Answer:"*, without any constrain on model's output space. However, for ScienceQA, we constrain the model's output to possible options (instead of open-ended), choose the option with highest confidence as model's prediction, and report the Top-1 accuracy.

The overall performance on image caption and general VQA tasks are reported in Table 4. As the results shown, our Qwen-VL and Qwen-VL-Chat both achieve obviously better results compared to previous generalist models in terms of both two tasks. Specifically, on zero-shot image caption task, Qwen-VL achieves state-of-the-art performance (*i.e.*, 85.8 CIDEr score) on the Flickr30K karpathy-test split, even outperforms previous generalist models with much more parameters (*e.g.*, Flamingo-80B with 80B parameters).

On general VQA benchmarks, our models also exhibit distinct advantages compared to others. On VQAv2, OKVQA and GQA benchmarks, Qwen-VL achieves 79.5, 58.6 and 59.3 accuracy respectively, which surpasses recent proposed LVLMs by a large margin. It's worth noting that Qwen-VL also shows strong zero-shot performance on ScienceQA and VizWiz datasets.

## 4.2 TEXT-ORIENTED VISUAL QUESTION ANSWERING

Text-oriented visual understanding has a broad application prospect in real-world scenarios. We assess our models' ability toward text-oriented visual question answering on several benchmarks including TextVQA (Sidorov et al., 2020), DocVQA (Mathew et al., 2021), ChartQA (Masry et al., 2022), AI2Diagram (Kembhavi et al., 2016), and OCR-VQA (Mishra et al., 2019). Similarly, the results are shown in Table 5. Compared to previous generalist models and recent LVLMs, our models show better performance on most benchmarks, frequently by a large margin.

## 4.3 REFER EXPRESSION COMPREHENSION

We show our models' fine-grained image understanding and localization ability by evaluating on a sort of refer expression comprehension benchmarks such as RefCOCO (Kazemzadeh et al., 2014), RefCOCOg (Mao et al., 2016), RefCOCO+ (Mao et al., 2016) and GRIT (Gupta et al., 2022). Specifically, the refer expression comprehension task requires the model to localize the target object under the guidance of a description. The results are shown in Table 6. Compared to previous generalist models or recent LVLMs, our models obtain top-tier results on all benchmarks.

## 4.4 FEW-SHOT LEARNING ON VISION-LANGUAGE TASKS

Our model also exhibits satisfactory in-context learning (*a.k.a.*, few-shot learning) ability. As shown in Figure 4, Qwen-VL achieves better performance through in-context few-shot learning on OKVQA (Marino et al., 2019), Vizwiz (Gurari et al., 2018), TextVQA (Sidorov et al., 2020), and Flickr30k (Young et al., 2014) when compared with models with similar number of parameters (Flamingo-9B(Alayrac et al., 2022), OpenFlamingo-9B(Awadalla et al., 2023) and IDEFICS-9BLaurençon et al. (2023)). Qwen-VL's performance is even comparable with much larger models (Flamingo-80B and IDEFICS-80B). Note that we adopt naïve random sample to construct the few-shot exemplars, sophisticated few-shot exemplar construction methods such as RICES (Yang et al., 2022b) are not used despite better results would be achieved.

Table 6: Results on Referring Expression Comprehension task.

| Model type | Model | RefCOCO | | | RefCOCO+ | | | RefCOCOg | | GRIT |
|---|---|---|---|---|---|---|---|---|---|---|
| | | val | test-A | test-B | val | test-A | test-B | val | test | refexp |
| Generalist Models | GPV-2 | - | - | - | - | - | - | - | - | 51.50 |
| | OFA-L* | 79.96 | 83.67 | 76.39 | 68.29 | 76.00 | 61.75 | 67.57 | 67.58 | 61.70 |
| | Unified-IO | - | - | - | - | - | - | - | - | **78.61** |
| | VisionLLM-H | | 86.70 | - | - | - | - | - | - | - |
| | Shikra-7B | 87.01 | 90.61 | 80.24 | 81.60 | 87.36 | 72.12 | 82.27 | 82.19 | 69.34 |
| | Shikra-13B | 87.83 | 91.11 | 81.81 | 82.89 | 87.79 | 74.41 | 82.64 | 83.16 | 69.03 |
| | **Qwen-VL-7B** | **89.36** | 92.26 | **85.34** | **83.12** | 88.25 | **77.21** | 85.58 | 85.48 | 78.22 |
| | **Qwen-VL-7B-Chat** | 88.55 | **92.27** | 84.51 | 82.82 | **88.59** | 76.79 | **85.96** | **86.32** | - |
| Specialist SOTAs | G-DINO-L | 90.56 | 93.19 | 88.24 | 82.75 | 88.95 | 75.92 | 86.13 | 87.02 | - |
| | UNINEXT-H | 92.64 | 94.33 | 91.46 | 85.24 | 89.63 | 79.79 | 88.73 | 89.37 | - |
| | ONE-PEACE | 92.58 | 94.18 | 89.26 | 88.77 | 92.21 | 83.23 | 89.22 | 89.27 | - |

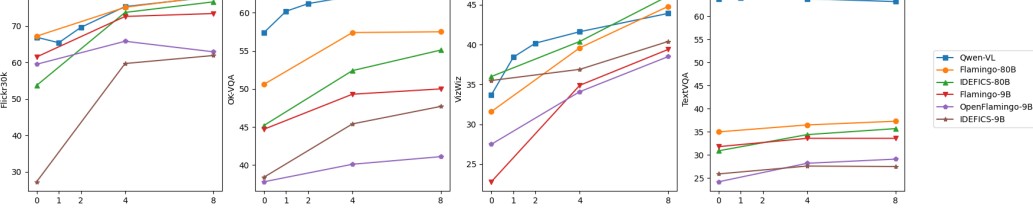

Figure 4: Few-shot learning results of Qwen-VL in comparison with other models.

## 4.5 INSTRUCTION FOLLOWING IN REAL-WORLD USER BEHAVIOR

In addition to previous conventional vision-language evaluations, to evaluate our Qwen-VL-Chat model's capacity under real-world user behavior, we further conduct the evaluations on the Touch-Stone (Bai et al., 2023), SEED-Bench (Li et al., 2023b), and MME (Fu et al., 2023). TouchStone is an open-ended vision-language instruction-following benchmark. We compare the instruction-following ability of Qwen-VL-Chat with other instruction-tuned LVLMs in both English and Chinese on the TouchStone benchmark. SEED-Bench consists of 19K multiple-choice questions with accurate human annotations for evaluating Multimodal LLMs, covering 12 evaluation dimensions including both the spatial and temporal understanding. MME measures both perception and cognition abilities on a total of 14 subtasks.

The results on three benchmarks are shown in Table 7. Qwen-VL-Chat has achieved obvious advantages over other LVLMs on all three datasets, indicating that our model performs better in understanding and answering diverse user instructions. In SEED-Bench, we have found that our model's visual capabilities can be effectively transferred to video tasks by simply sampling four frames. In terms of the overall scores presented in TouchStone, our model demonstrates a clear advantage compared to other LVLMs, especially in terms of its Chinese capabilities. In terms of the broad categories of abilities, our model exhibits a more pronounced advantage in understanding and recognition, particularly in areas such as text recognition and chart analysis. For more detailed information, please refer to the TouchStone dataset.

## 5 RELATED WORK

In recent years, researchers have shown considerable interest in vision-language learning (Su et al., 2019; Chen et al., 2020; Li et al., 2020; Zhang et al., 2021; Li et al., 2021b; Lin et al., 2021; Kim et al., 2021; Dou et al., 2022; Zeng et al., 2021; Li et al., 2021a; 2022), especially in the development of multi-task generalist models (Hu & Singh, 2021; Singh et al., 2022; Zhu et al., 2022; Yu et al., 2022; Wang et al., 2022a; Lu et al., 2022a; Bai et al., 2022). CoCa (Yu et al., 2022) proposes an encoder-decoder structure to address image-text retrieval and vision-language genera-

Table 7: Results on Instruction-following benchmarks.

| Model | TouchStone | | SEED-Bench | | | MME | |
| | En | Cn | All | Img | Video | Perception | Cognition |
|---|---|---|---|---|---|---|---|
| VisualGLM | - | 247.1 | - | - | - | 705.31 | 181.79 |
| PandaGPT | 488.5 | - | - | - | - | 642.59 | 228.57 |
| MiniGPT4 | 531.7 | - | 42.8 | 47.4 | 29.9 | 581.67 | 144.29 |
| InstructBLIP | 552.4 | - | 53.4 | 58.8 | 38.1 | 1212.82 | 291.79 |
| LLaMA-AdapterV2 | 590.1 | - | 32.7 | 35.2 | 25.8 | 972.67 | 248.93 |
| LLaVA | 602.7 | - | 33.5 | 37.0 | 23.8 | 502.82 | 214.64 |
| mPLUG-Owl | 605.4 | - | 34.0 | 37.9 | 23.0 | 967.34 | 276.07 |
| **Qwen-VL** | - | - | 56.3 | 62.3 | **39.1** | - | - |
| **Qwen-VL-Chat** | **645.2** | **401.2** | **58.2** | **65.4** | 37.8 | **1487.58** | **360.71** |

tion tasks simultaneously. OFA (Wang et al., 2022a) transforms specific vision-language tasks into sequence-to-sequence tasks using customized task instructions. Unified I/O (Lu et al., 2022a) further introduces more tasks like segmentation and depth estimation into a unified framework. Another category of research focuses on building vision-language representation models (Radford et al., 2021; Jia et al., 2021; Zhai et al., 2022; Yuan et al., 2021; Yang et al., 2022a). CLIP (Radford et al., 2021) leverages contrastive learning and large amounts of data to align images and language in a semantic space, resulting in strong generalization capabilities across a wide range of downstream tasks. BEIT-3 (Wang et al., 2022b) employs a mixture-of-experts (MOE) structure and unified masked token prediction objective, achieving state-of-the-art results on various visual-language tasks. In addition to vision-language learning, ImageBind (Girdhar et al., 2023) and ONE-PEACE (Wang et al., 2023) align more modalities such as speech into a unified semantic space, thus creating more general representation models.

With the rapid development of large language models (LLMs) (Brown et al., 2020; OpenAI, 2023; Anil et al., 2023; Gao et al., 2023; Qwen, 2023), researchers have started building more powerful large vision-language models (LVLMs) based on LLMs (Alayrac et al., 2022; Chen et al., 2022; Li et al., 2023c; Dai et al., 2023; Huang et al., 2023; Peng et al., 2023; Zhu et al., 2023; Liu et al., 2023; Ye et al., 2023b;a; Chen et al., 2023a; Li et al., 2023a; Zhang et al., 2023; Sun et al., 2023). BLIP-2 (Li et al., 2023c) proposes Q-Former to align the frozen vision foundation models and LLMs. Meanwhile, LLAVA (Liu et al., 2023) and Mini-GPT4 (Zhu et al., 2023) introduce visual instruction tuning to enhance instruction following capabilities in LVLMs. Additionally, mPLUG-DocOwl (Ye et al., 2023a) incorporates document understanding capabilities into LVLMs by introducing digital documents data. Kosmos2 (Peng et al., 2023), Shikra (Chen et al., 2023a), and BuboGPT (Zhao et al., 2023) further enhance LVLMs with visual grounding abilities, enabling region description and localization. Despite achieving significant progress, previous vision-language models still have several limitations such as poor robustness in instruction following, limited generalization capabilities in unseen tasks, and a lack of in-context abilities. To further explore the performance extremity and efficient training strategy in terms of both data organization and model optimization, we train our LVLMs on a vast of vision-language datasets with kinds of multimodal tasks (*e.g.*, image caption, visual question answering, document comprehension, and visual grounding.). The resulting models, Qwen-VL and Qwen-VL-chat, not only exhibit top-tier performance across a wide range conventional benchmarks, but also outperform previous vision-language models on several recently proposed instruction following benchmarks.

## 6 CONCLUSION AND FUTURE WORK

We release the Qwen-VL series, a set of large-scale multilingual vision-language models that aims to facilitate multimodal research. Qwen-VL outperforms similar models across various benchmarks, supporting multilingual conversations, multi-image interleaved conversations, grounding in Chinese, and fine-grained recognition. Moving forward, we are dedicated to further enhancing Qwen-VL's capabilities in several key dimensions: **(i)** Integrating Qwen-VL with more modalities, such as speech and video. **(ii)** Augmenting Qwen-VL by scaling up the model size, training data and higher resolution, enabling it to handle more complex and intricate relationships within multimodal data. **(iii)** Expanding Qwen-VL's prowess in multi-modal generation, specifically in generating high-fidelity images and fluent speech.

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

# A  DATASET DETAILS

## A.1  IMAGE-TEXT PAIRS

We use web-crawled image-text pairs dataset for pre-training, which includes LAION-en (Schuhmann et al., 2022a), LAION-zh (Schuhmann et al., 2022a), LAION-COCO (Schuhmann et al., 2022b), DataComp (Gadre et al., 2023) and Coyo (Byeon et al., 2022). We clean these noisy data by several steps:

1. Removing pairs with too large aspect ratio of the image
2. Removing pairs with too small image
3. Removing pairs with a harsh CLIP score (dataset-specific)
4. Removing pairs with text containing non-English or non-Chinese characters
5. Removing pairs with text containing emoji characters
6. Removing pairs with text length too short or too long
7. Cleaning the text's HTML-tagged part
8. Cleaning the text with certain unregular patterns

For academic caption datasets, we remove pairs whose text contains the special tags in CC12M (Changpinyo et al., 2021) and SBU (Ordonez et al., 2011). If there is more than one text matching the same image, we select the longest one.

## A.2  VQA

For the VQAv2 (Goyal et al., 2017) dataset, we select the answer annotation based on the maximum confidence. For other VQA datasets, we didn't do anything special.

## A.3  GROUNDING

For the GRIT (Peng et al., 2023) dataset, we found that there are many recursive grounding box labels in one caption. We use the greedy algorithm to clean the caption to make sure each image contains the most box labels with no recursive box labels. For other grounding datasets, we simply concatenate the noun/phrase with respective bounding box coordinates.

## A.4  OCR

We generated the synthetic OCR dataset using Synthdog (Kim et al., 2022). Specifically, we use the COCO (Lin et al., 2014) train2017 and unlabeld2017 dataset split as the natural scenery background. Then we selected 41 English fonts and 11 Chinese fonts to generate text. We use the default hyperparameters as in Synthdog. We track the generated text locations in the image and convert them to quadrilateral coordinates and we also use these coordinates as training labels. The visualization example is illustrated in the second row of Fig 5.

For all the PDF data we collected, we follow the steps below to pre-process the data using PyMuPDF (Software, 2015) to get the rendering results of each page in a PDF file as well as all the text annotations with their bounding boxes.

1. Extracting all texts and their bounding boxes for each page.
2. Rendering each page and save them as an image file.
3. Removing too small image.
4. Removing images with too many or too few characters.
5. Removing images containing Unicode characters in the "Latin Extended-A" and "Latin Extended-B" blocks.
6. Removing images containing Unicode characters in the "Private Use Area (PUA)" block.

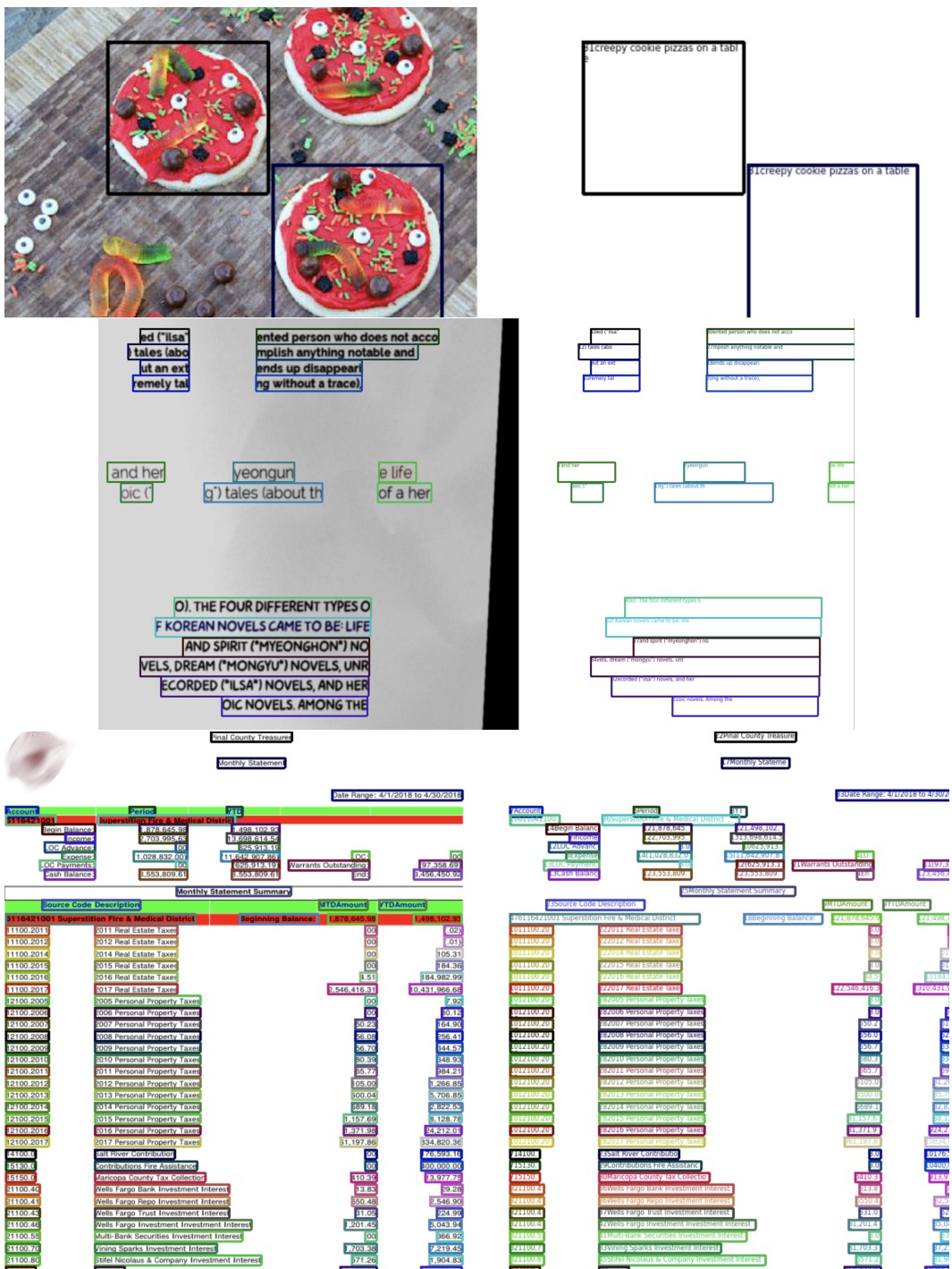

Figure 5: Visualization of the Grounding and OCR data used for training Qwen-VL

For all HTML web pages we collected, we pre-process them in a similar approach to all the PDF data we collected, but we use Puppeteer (Google, 2023) instead of PyMuPDF to render these HTML pages and get the ground truth annotation. We follow the steps below to pre-process the data.

1. Extracting all texts for each webpage.

2. Rendering each page and save them as an image file.

3. Removing too small image.

4. Removing images with too many or too few characters.

5. Removing images containing Unicode characters in the "Private Use Area (PUA)" block.

# B   DATA FORMAT DETAILS OF TRAINING

## B.1   DATA FORMAT OF MULTI-TASK PRE-TRAINING

We visualize the Multi-Task Pre-training data format in Box B.1. The Box contains all 7 tasks with the black-colored text as the prefix sequence without loss and blue-colored text as the ground truth labels with loss.

---

**Image Captioning**

cc3m/01581435.jpg</img>Generate the caption in English: the beautiful flowers for design.<eos>

**Vision Question Answering**

VG_100K_2/1.jpg</img> Does the bandage have a different color than the wrist band? Answer: No, both the bandage and the wrist band are white.<eos>

**OCR VQA**

ocr_vqa/1.jpg</img> What is the title of this book? Answer: Asi Se Dice!, Volume 2: Workbook And Audio Activities (Glencoe Spanish) (Spanish Edition)<eos>

**Caption with Grounding**

coyo700m/1.jpg</img>Generate the caption in English with grounding: Beautiful shot of <ref>bees</ref><box>(661,612),(833,812)</box><box>(120,555),(265,770)</box> gathering nectars from <ref>an apricot flower</ref><box>(224,13),(399,313)</box><eos>

**Referring Grounding**

VG_100K_2/3.jpg</img><ref>the ear on a giraffe</ref><box>(176,106),(232,160)</box><eos>

**Grounded Captioning**

VG_100K_2/4.jpg</img><ref>This</ref><box>(360,542),(476,705)</box> is Yellow cross country ski racing gloves<eos>

**OCR**

synthdog/1.jpg</img>OCR with grounding: <ref>It is managed</ref> <quad>(568,121), (625,131), (624,182), (567,172)</quad>...<eos>

---

## B.2   DATA FORMAT OF SUPERVISED FINE-TUNING

To better accommodate multi-image dialogue and multiple image inputs, we add the string "Picture $id$:" before different images, where the $id$ corresponds to the order of image input dialogue. In terms of dialogue format, we construct our instruction tuning dataset using the ChatML (Openai) format, where each interaction's statement is marked with two special tokens (<im_start> and <im_end>) to facilitate dialogue termination.

> **The Dataset Format Example of ChatML**
>
> <im_start>user
> Picture 1: vg/VG_100K_2/649.jpg</img>What is the sign in the picture?<im_end>
> <im_start>assistant
> The sign is a road closure with an orange rhombus.<im_end>
> <im_start>user
> How is the weather in the picture?<im_end>
> <im_start>assistant
> The shape of the road closure sign is an orange rhombus.<im_end>

During training, we ensure the consistency between prediction and training distributions by only supervising answers and special tokens (blue in the example), and not supervising role names or question prompts.

## C  HYPERPARAMETERS

We report the detailed training hyperparameter settings of Qwen-VL in Table 8.

Table 8: Training hyperparameters of Qwen-VL

| Configuration | Pre-training | Multi-task Pre-training | Supervised Fine-tuning |
|---|---|---|---|
| ViT init. | Open-CLIP-bigG | Qwen-VL 1st-stage | Qwen-VL 2nd-stage |
| LLM init. | Qwen-7B | Qwen-7B | Qwen-VL 2nd-stage |
| VL Adapter init. | random | Qwen-VL 1st-stage | Qwen-VL 2nd-stage |
| Image resolution | $224^2$ | $448^2$ | $448^2$ |
| ViT sequence length | 256 | 1024 | 1024 |
| LLM sequence length | 512 | 2048 | 2048 |
| Learnable query numbers | 256 | 256 | 256 |
| Optimizer | | AdamW | |
| Optimizer hyperparameter | | $\beta_1 = 0.9, \beta_2 = 0.98, eps = 1e^{-6}$ | |
| Peak learning rate | $2e^{-4}$ | $5e^{-5}$ | $1e^{-5}$ |
| Minimum learning rate | $1e^{-6}$ | $1e^{-5}$ | $1e^{-6}$ |
| ViT learning rate decay | 0.95 | 0.95 | 0 |
| ViT Drop path rate | | 0 | |
| Learning rate schedule | | cosine decay | |
| Weight decay | | 0.05 | |
| Gradient clip | | 1.0 | |
| Training steps | 50k | 19k | 8k |
| Warm-up steps | 500 | 400 | 3k |
| Global batch size | 30720 | 4096 | 128 |
| Gradient Acc. | 6 | 8 | 8 |
| Numerical precision | | `bfloat16` | |
| Optimizer sharding | | ✓ | |
| Activation checkpointing | | ✗ | |
| Model parallelism | ✗ | 2 | 2 |
| Pipeline parallelism | | ✗ | |

In the first pre-training stage, the model is trained using AdamW optimizer with $\beta_1 = 0.9, \beta_2 = 0.98, eps = 1e^{-6}$. We use the cosine learning rate schedule and set the maximum learning rate of $2e^{-4}$ and minimum of $1e^{-6}$ with a linear warm-up of 500 steps. We use a weight decay of $5e^{-2}$ and a gradient clipping of 1.0. For the ViT image encoder, we apply a layer-wise learning rate decay

strategy with a decay factor of $0.95$. The training process uses a batch size of 30720 for the image-text pairs, and the entire first stage of pre-training lasts for 50,000 steps, consuming approximately 1.5 billion image-text samples and 500 billion image-text tokens.

In the second multi-task training stage, we increase the input resolution of the visual encoder from $224 \times 224$ to $448 \times 448$, reducing the information loss caused by image down-sampling. We unlocked the large language model and trained the whole model. The training objective is the same as the pre-training stage. We use AdamW optimizer with $\beta_1 = 0.9, \beta_2 = 0.98, eps = 1e^{-6}$. We trained for 19000 steps with 400 warm-up steps and a cosine learning rate schedule. Specifically, we use the model parallelism techniques for ViT and LLM.

## D  SUMMARY OF THE EVALUATION BENCHMARKS

We provide a detailed summary of the used evaluation benchmarks and corresponding metrics in Table 9.

Table 9: Summary of the evaluation benchmarks.

| Task | Dataset | Description | Split | Metric |
|---|---|---|---|---|
| Image Caption | Nocaps | Captioning of natural images | val | CIDEr($\uparrow$) |
| | Flickr30K | Captioning of natural images | karpathy-test | CIDEr($\uparrow$) |
| General VQA | VQAv2 | VQA on natural images | test-dev | VQA Score($\uparrow$) |
| | OKVQA | VQA on natural images requiring outside knowledge | val | VQA Score($\uparrow$) |
| | GQA | VQA on scene understanding and reasoning | test-balanced | EM($\uparrow$) |
| | ScienceQA-Img | Multi-choice VQA on a diverse set of science topics | test | Accuracy($\uparrow$) |
| | VizWiz | VQA on photos taken by people who are blind | test-dev | VQA Score($\uparrow$) |
| Text-oriented VQA | TextVQA | VQA on natural images containing text | val | VQA Score($\uparrow$) |
| | DocVQA | VQA on images of scanned documents | test | ANLS($\uparrow$) |
| | ChartQA | VQA on images of charts | test | Relaxed EM($\uparrow$) |
| | OCRVQA | VQA on images of book covers | test | EM($\uparrow$) |
| | AI2Diagram | VQA on images of scientific diagrams | test | EM($\uparrow$) |
| Refer Expression Comprehension | RefCOCO | Refer grounding on natural images | val & testA & testB | Accuracy($\uparrow$) |
| | RefCOCO+ | Refer grounding on natural images | val & testA & testB | Accuracy($\uparrow$) |
| | RefCOCOg | Refer grounding on natural images | val & test | Accuracy($\uparrow$) |
| | GRiT | Refer grounding on natural images | test | Accuracy($\uparrow$) |
| Instruction Following | TouchStone | Open-ended VL instruction following benchmark | English & Chinese | GPT-4 Score ($\uparrow$) |
| | MME | Open-ended VL Benchmark by yes/no questions | Perception & Cognition | Accuracy ($\uparrow$) |
| | Seed-Bench | Open-ended VL Benchmark by Multi-choice VQA | Image & Video | Accuracy ($\uparrow$) |

## E  ADDITIONAL EXPERIMENTAL DETAILS

### E.1  CONVERGENCE OF THE PRE-TRAINING STAGE

In Figure 6, we show the convergence of the Pre-training Stage (stage one). The whole models are trained using BFloat16 mixed precision, the batch size is 30720, and the learning rate is $2e^{-4}$. All images are only trained once (one epoch). The training loss decreases steadily with the increase of the number of training pictures. Note that, the pre-training stage (Stage one) has no VQA data being added, but the Zero-shot VQA score increases amidst fluctuations.

### E.2  NUMBER OF LEARNABLE QUERIES IN THE VISION-LANGUAGE ADAPTER

The vision-language adapter uses cross-attention to compress the visual feature sequence by a set of learning queries of length. Too few queries can lead to the loss of some visual information, while too many queries may reduce in greater convergence difficulty and computational cost.

An ablation experiment is conducted on the number of learnable queries in the vision-language adapter. We used ViT-L/14 as the visual encoder and the $224 \times 224$ resolution picture as input, so the sequence length of ViT's output is $(224/14)^2 = 256$. As shown in the left part of Figure 7, the fewer queries used at the beginning of training, the lower the initial loss. However, with convergence, too many or too few queries will cause convergence to slow down, as shown in the right part of Figure

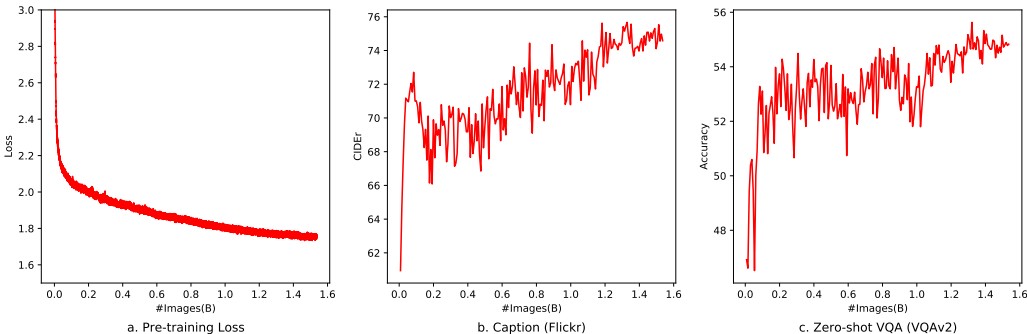

Figure 6: Visualization of the Convergence of the Pre-training Stage

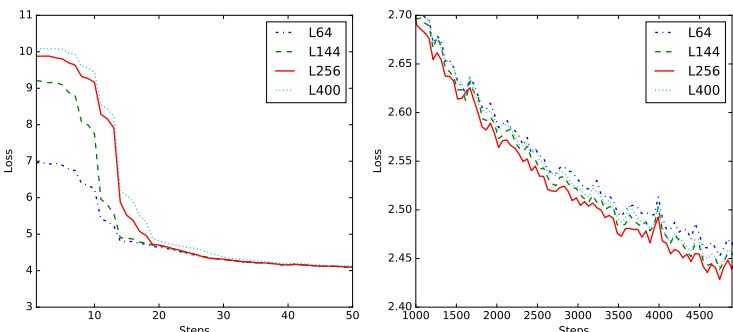

Figure 7: Visualization of the loss in pre-training stage (stage 1) when using different compressed feature lengths of the vision-language adapter. The left depicts the initial training loss (within 50 steps), and the right depicts the loss in convergence (1k-5k steps). In the legend, L64 denotes that the adapter uses 64 queries to compress the visual feature sequence to a fixed length of 64, and so on. The loss curves have been smoothed to avoid shading owing to fluctuations.

7. Considering that the second training stage (Multi-task Pre-train) applies 448*448 resolution, where the sequence length of ViT's output is $(448/14)^2 = 1024$. Too few queries can result in more information being lost. We finally chose to use 256 queries for the vision-language adapter in Qwen-VL.

### E.3 WINDOW ATTENTION VS GLOBAL ATTENTION FOR VISION TRANSFORMER

Using a high-resolution Vision Transformer in the model will significantly increase the computational cost. One possible solution to reduce the computational cost of the model is to use Window Attention in the Vision Transformer, *i.e.*, to perform Attention only in a window of $224 \times 224$ in most layers of the ViT part of the model, and to perform Attention for the full $448 \times 448$ or $896 \times 896$ image in a small number of layers (e.g. 1 out of every 4 layers) of the ViT part of the model.

To this end, we conducted ablation experiments to compare the performance of the model when using Global Attention and Window Attention for ViT. We compare the experimental results for analysing the trade-off between computational efficiency and convergence of the model.

As shown in Figure 8 and Table 10, the loss of the model is significantly higher when Window Attention instead of Vanilla Attention is used. And the training speeds for both of them are similar. Therefore, we decided to use Vanilla Attention instead of Window Attention for the Vision Transformer when training Qwen-VL.

The reason we don't use Window Attention with $896 \times 896$ resolution is that its training speed is too slow for us. Although it reaches a loss value similar to model with $448 \times 448$ resolution input at 5000 steps. It takes almost 2.5 times longer to train than the model with $448 \times 448$ resolution input.

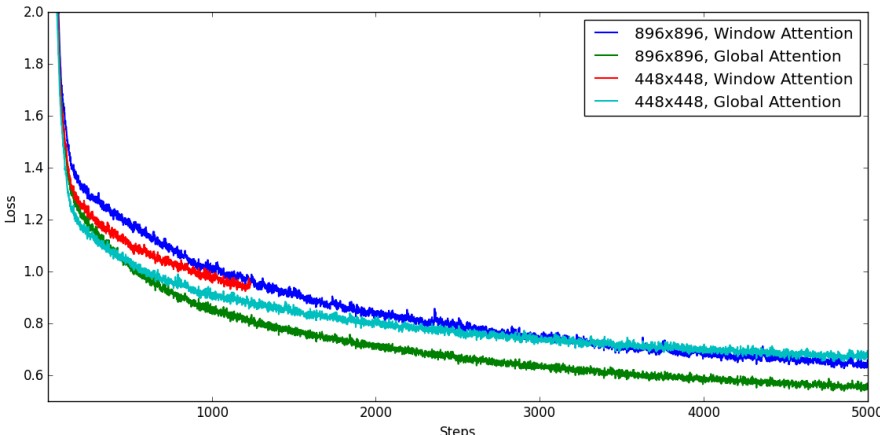

Figure 8: Visualization of the Loss when using Window Attention vs Global Attention

Table 10: Training speed of Window Attention vs Global Attention for different input image resolutions

| Model input resolution & Attention type | Training speed |
|---|---|
| $448 \times 448$, Global Attention | 10s / iter |
| $448 \times 448$, Window Attention | 9s / iter |
| $896 \times 896$, Global Attention | 60s / iter |
| $896 \times 896$, Window Attention | 25s / iter |

### E.4 PERFORMANCE ON PURE-TEXT TASKS

In order to study the effect of multi-modal training on pure-text ability, we show the performance of pure-text tasks of Qwen-VL compared to open-source LLM in Table 11.

Qwen-VL uses an intermediate checkpoint of Qwen-7B as the LLM initialization. The reason why we did not use the final released checkpoint of Qwen-7B is that Qwen-VL and Qwen-7B were developed at a very similar period. Because Qwen-VL has a good initialization on LLM by Qwen-7B, it is comparable to many text-only LLMs on pure-text tasks.

Table 11: Performance on Pure-text Benchmarks of Qwen-VL compared to open-source LLM. Due to the introduction of pure-text data in the multi-task training and SFT stage, Qwen-VL do not compromise any pure-text ability.

| Model | MMLU | CMMLU | C-Eval |
|---|---|---|---|
| LLaMA-7B | 35.1 | 26.8 | - |
| LLaMA2-7B | 46.8 | 31.8 | 32.5 |
| Baichuan-7B | 42.3 | 44.4 | 42.8 |
| Baichuan2-7B | 54.2 | 57.1 | 54.0 |
| ChatGLM2-6B | 47.9 | 48.8 | 51.7 |
| InternLM-7B | 51.0 | 51.8 | 52.8 |
| Qwen-7B (final released) | 58.2 | 62.2 | 63.5 |
| Qwen-7B (intermediate, use as Qwen-VL's LLM initialization) | 49.9 | - | 48.5 |
| Qwen-VL | 50.7 | 49.5 | 51.1 |

Furthermore, in the multi-task training and SFT stages, Qwen-VL not only utilizes visual and language-related data but also incorporates pure-text data for training. The purpose of this is to prevent the catastrophic forgetting of text comprehension by leveraging the information from pure-text data. The results in Table 11 indicate that the Qwen-VL model does not exhibit any degradation in terms of its pure text capability and even demonstrates improvement after multi-task training.

## COMPARISONS WITH OTHER OPEN-SOURCE LVLMs

In Tab. 12, we compare Qwen-VL with previous open-source large vision-language models (LVLMs) in terms of their supporting language and capable task. As shown in the table, Qwen-VL can support both English and Chinese in its application and is capable to finish four kinds of vision-language tasks within the same model. Moreover, as aforementioned, Qwen-VL also outperforms these predecessors across several benchmarks on these four tasks.

Table: 12: Comparisons between Qwen-VL and previous open-source LVLMs in terms of both language support and task capacity. For text-oriented VQA, we refer to whether the model is designed or optimized to tackle this problem explicitly. Grounding indicates visual grounding.

| Model | Language | | Task | | | |
| | English | Chinese | Caption | General VQA | Text-oriented VQA | Grounding |
|---|---|---|---|---|---|---|
| Kosmos | ✓ | | ✓ | ✓ | | |
| BLIP2 | ✓ | | ✓ | ✓ | | |
| LLaVA | ✓ | | ✓ | ✓ | | |
| MiniGPT-4 | ✓ | | ✓ | ✓ | | |
| ChatGLM | ✓ | ✓ | ✓ | ✓ | | |
| mPLUG-Owl | ✓ | | ✓ | ✓ | | |
| InstructBLIP | ✓ | | ✓ | ✓ | ✓ | |
| mPLUG-DocOwl | ✓ | | ✓ | ✓ | ✓ | |
| Kosmos2 | ✓ | | ✓ | ✓ | | ✓ |
| Shikra | ✓ | | ✓ | ✓ | | ✓ |
| Qwen-VL | ✓ | ✓ | ✓ | ✓ | ✓ | ✓ |

## ABLATION ON IMAGE CONTAMINATION

To verify the potential negative impact of image contamination (especially between COCO-based datasets and benchmarks) on our Qwen-VL's final performance, we conduct two experiments and evaluate two models across a wide range of benchmarks. Specifically, we employe a LLM with 1.8B parameters and CLIP-ViT-L/14 as vision encoder, the overall parameters in our model are about 2B. We train this model on the same dataset composition as we used in our paper, in spite of much fewer iterations. During the first stage, we tune the model with a batch size of 30720 for 8000 steps. During the second stage, we train both models for 6000 steps with a batch size of 2048. For the baseline experiment, we use exact the same composition of datasets as in our paper. For the second experiment, we follow previous work to perform near-deduplication. Specifically, we deduplicate all coco images used in some coco-based evaluation benchmarks (e.g., VQA, OKVQA, Refcoco/Refcoco+/Refcocog). The results are shown in Tab. 13.

## ABLATION ON THE TRAINING PART IN SECOND STAGE

We conduct additional ablation studies on the training part during the second stage. Specifically, we employe a LLM with 1.8B parameters and CLIP-ViT-L/14 as vision encoder, the overall parameters in our model are about 2B. During the first stage, we tune the model with a batch size of 30720 for 8000 steps. During the second stage, in addition to the default setup, we also try to train our model with either ViT or LLM being frozen. All these three models for the second stage are tuned for 6000 steps with a batch size of 2048.

Table: 13: We conduct contrast experiments to verify the effect of potential image contamination on the final results or Qwen-VL. As the results shown, with COCO-based image carefully deduplicated in both the first and the second training stages, we observe little to no performance perturbation between two experiments for both coco-based and non coco-based benchmarks.

| Benchmark | Is COCO-based? | Qwen-VL(2B) | w/. Image Deduplication |
|---|---|---|---|
| Nocaps(val) | | **114.7** | 114.3 |
| VQAv2(val) | ✓ | **74.6** | 74.3 |
| OKVQA(val) | ✓ | 45.7 | **45.9** |
| VizWiz(val) | | 29.4 | **29.8** |
| TextVQA(val) | | **52.5** | **52.5** |
| RefCOCO(val) | ✓ | **79.9** | 79.6 |
| RefCOCO+(val) | ✓ | 66.1 | **66.5** |
| RefCOCOg(val) | ✓ | **74.2** | 73.6 |

As the results in Tab. 14. There are several observations:

- Compared to jointly tuning both ViT and LLM, freeze ViT or freeze LLM both lead to significant performance decrease.
- On most benchmarks, freeze ViT shows better results than freeze LLM. This observation is in line with some current vision-language machines who prefer to tune LLM and keep the ViT frozen (Huang et al., 2023; Chen et al., 2023a).
- However, for text-oriented VQA(*i.e.*, TextVQA), it's interesting to see that freeze LLM (tune ViT) shows significant higher accuracy compared to freeze ViT (tune LLM). We ascribe this phenomenon to text-oriented VQA requires more fine-grained visual perception capacity which has not been learnt in previous ViT training process(*e.g.*, vision-language contrastive learning in CLIP, and coarse-grained image captioning in our first stage).

Table: 14: Ablation studies on the training part in second stage.

| Benchmark | Qwen-VL(2B) | Freeze ViT in Stage2 | Freeze LLM in Stage2 |
|---|---|---|---|
| Nocaps(val) | 114.7 | 109.7 | 99.3 |
| VQAv2(val) | 74.6 | 68.9 | 64.4 |
| OKVQA(val) | 45.7 | 43.2 | 32.7 |
| VizWiz(val) | 29.4 | 22.5 | 19.3 |
| TextVQA(val) | 52.5 | 27.7 | 40.1 |

MODEL FAIRNESS, BIAS, AND OTHER POTENTIAL ISSUES

Following previous vision-language models Chen et al. (2023b), we evaluate the overall level of toxicity and profanity of Qwen-VL's generated captions on FairFace dataset Kärkkäinen & Joo (2019). Specifically, we instruct our model to generate descriptions for each image in FairFace dataset Kärkkäinen & Joo (2019) val split, then the generated captions are scored by Perspective API Lees et al. (2022) for their degree of toxicity and profanity. Tables 15- 17 depict the percentage of varying degree of toxicity or profanity splitted by subgroups related to race, age, and gender, respectively. Our observations are: **(i)** Qwen-VL shows a general low degree of toxicity across all slices, and an extremly low level of profanity (more than 99% captions' profanity are below 0.2). **(ii)** Taking 0.8 as threshold for both toxicity and profanity, Qwen-VL's outputs are under great satefy (not a single caption is toxic nor profane). **(iii)** As shown in Tab. 18, compared to previous method, *i.e.*, PaLI-X (Chen et al., 2023b), Qwen-VL is much more safe across all slices.

Table: 15: Toxicity and profanity of Qwen-VL on different subgroups related to race.

| RACE | Toxicity < 0.2 | Toxicity 0.2 - 0.8 | Toxicity > 0.8 | Profanity < 0.2 | Profanity 0.2 - 0.8 | Profanity > 0.8 |
|---|---|---|---|---|---|---|
| Black | 84.1% | 15.9% | 0% | 99.7% | 0.3% | 0% |
| East Asian | 86.2% | 13.8% | 0% | 99.2% | 0.8% | 0% |
| Indian | 80.5% | 19.5% | 0% | 99.7% | 0.3% | 0% |
| Latino Hispanic | 86.3% | 13.7% | 0% | 99.3% | 0.7% | 0% |
| Middle Eastern | 86.4% | 13.6% | 0% | 99.3% | 0.7% | 0% |
| Southeast Asian | 86.7% | 13.3% | 0% | 99.2% | 0.8% | 0% |
| White | 84.9% | 15.1% | 0% | 99% | 1% | 0% |

Table: 16: Toxicity and profanity of Qwen-VL on different subgroups related to age.

| AGE | Toxicity < 0.2 | Toxicity 0.2 - 0.8 | Toxicity > 0.8 | Profanity < 0.2 | Profanity 0.2 - 0.8 | Profanity > 0.8 |
|---|---|---|---|---|---|---|
| 0-2 | 68.8% | 31.2% | 0% | 100% | 0% | 0% |
| 3-9 | 83.4% | 16.6% | 0% | 99% | 1% | 0% |
| 10-19 | 86.4% | 13.6% | 0% | 99.2% | 0.8% | 0% |
| 20-29 | 85.9% | 14.1% | 0% | 99.4% | 0.6% | 0% |
| 30-39 | 85.5% | 14.5% | 0% | 99.4% | 0.6% | 0% |
| 40-49 | 85.7% | 14.3% | 0% | 99.2% | 0.8% | 0% |
| 50-59 | 83.4% | 16.6% | 0% | 99.6% | 0.4% | 0% |
| 60-69 | 86.6% | 13.4% | 0% | 99.7% | 0.3% | 0% |
| >= 70 | 75.4% | 24.6% | 0% | 99.2% | 0.8% | 0% |

Table: 17: Toxicity and profanity of Qwen-VL on different subgroups related to gender.

| GENDER | Toxicity < 0.2 | Toxicity 0.2 - 0.8 | Toxicity > 0.8 | Profanity < 0.2 | Profanity 0.2 - 0.8 | Profanity > 0.8 |
|---|---|---|---|---|---|---|
| Male | 86% | 14% | 0% | 99.6% | 0.4% | 0% |
| Female | 83.8% | 16.2% | 0% | 99.1% | 0.9% | 0% |

Table: 18: Comparision between Qwen-VL and PaLI-X Chen et al. (2023b).

| MODEL | | Threshold | Race (Low/High) (↓) | Age (Low/High) (↓) | Gender (Low/High) (↓) |
|---|---|---|---|---|---|
| PaLI-X | Toxicity | 0.2 | 35.8%/40.4% | 33.9%/40.0% | - |
| Qwen-VL | Toxicity | 0.2 | **13.3%/19.5%** | **13.4%/31.2%** | 14.0%/16.2% |
| PaLI-X | Profanity | 0.2 | 5.1%/8.5% | 3.5%/10.3% | - |
| Qwen-VL | Profanity | 0.2 | **0.3%/1.0%** | **0.3%/1.0%** | 0.4%/0.9% |

