# OpenReview forum: "Qwen-VL: A Versatile Vision-Language Model for Understanding, Localization, Text Reading, and Beyond"
_ICLR.cc/2024/Conference — Submitted to ICLR 2024_

### Official Review · Reviewer_2REz · 2023-11-01

**Soundness:** 3 good
**Presentation:** 2 fair
**Contribution:** 2 fair
**Rating:** 5
**Confidence:** 4

**Summary:**

This model introduced a Vision Language Model QWEN-VL which is both pretrained and instruction finetuned. The model shows decent multimodal capability, especially in terms of bounding box reasoning. The model will be open sourced which will be helpful to the community.

**Strengths:**

- Open sourcing the model is going to help the research community
- The model shows decent multimodal capability, especially in terms of bounding box reasoning

**Weaknesses:**

My concerns are regarding the scientific and technical contributions from this paper.
- The claim in the related work section, "Despite achieving significant progress, previous vision-language models still have several limitations such as poor robustness in instruction following, limited generalization capabilities in unseen tasks, and a lack of in-context abilities." lacks justification. For example many models are not instruction-tuned (yet). That does not mean they have a fundamental difficulty in instruction following.
- There is limited innovation on the model architecture and training recipe. For example the use of interleaved data and multi-stage, multi-resolution training has been proposed in previous works. Also there is limited novelty in showing that supervised finetuning with interleaved chat data can lead to chatting capability.
- The ablation study is not written clearly (also see questions below)

**Questions:**

In the ablation study of Figure 7, which stage is that, lower-res or higher-res? If it is lower-res (224) stage it makes sense to use 256 tokens as the native number of patches is just (224/14)^2 = 256. If it is the higher-res (448) stage, then it is counterintuitive that using more tokens, i.e., 400, with more degrees of freedom, will lead to worse performance.

**Details Of Ethics Concerns:**

Since it is an open source generative model, there should be analysis regarding Discrimination / bias / fairness

---

> ### Author Response · Authors · 2023-11-20
> **Response to Reviewer 2REz (1/2)**
>
> **[Related work clarity]**
>
> This is a missunderstanding, the limitations we mentioned were not intended to challenge previous vision-language models for lacking instruction tuning process. On the other hand, we meant to pinpoint the potential improving direction compared to existing vision-language chatbots (*e.g.*, VisualGLM, PandaGPT, MiniGPT4, and etc.). And as the results shown in Table 7, compared to these predecessors, Qwen-VL exhibits better instruction following ability on three recently proposed instruction following evaluation benchmarks.
>
> *For better readability, we have updated this section in the revision.*
>
> **[Innovation on model architecture and recipe]**
>
> We thank the reviewer's valuable feedback. We restate Qwen-VL's contribution from several distinct perspectives in the general response to all reviewers. We hope that our detailed reply will ease your concerns.
>
> **[Ablation study details]**
>
> Thanks for pointing this out! The experiments in Figure $7$ are all conducted in the **fist** stage, with $224\times224$ resolution for image inputs. That is to say, our vision encoder will pathify each input image into $256$ patches, and under this circumstance we found that inadequate and superfluous numbers of queries both lead to slightly higher loss in the later training period.
>
> *We have updated this part in the revision to make it more clear.*

---

> ### Author Response · Authors · 2023-11-20
> **Response to Reviewer 2REz (2/2)**
>
> **[Analysis on model fairness, biases and other potential issues]**
>
> We thank the reviewer for the suggestion. We acknowledge that in-depth analysises on models' fairness, biases and other potential issues can largely help the communities better understand our models and should be considered as an important part of our open source project. In this light, we follow previous vision-language models [1, 2] and evaluate the overall level of toxicity and profanity of Qwen-VL's generated captions.
>
> We instruct Qwen-VL to generate descriptions for each image in FairFace dataset [3] val split, then the generated captions are scored by Perspective API [1] for their degree of toxicity and profanity. Three tables below depict the percentage of varying degree of toxicity or profanity splitted by subgroups related to race, age, and gender, respectively. The conclusions are:
>
> * Qwen-VL shows a general low degree of toxicity across all slices, and an extremly low level of profanity (more than $99$% captions' profanity are below $0.2$).
> * Taking $0.8$ as a threshold for both toxicity and profanity, Qwen-VL's outputs are under great satefy (not a single caption is toxic nor profane).
> * Compared to previous method, *i.e.*, PaLI-X, Qwen-VL is much more safe across all slices.
>
> The last table compare our model with PaLI-X on the degree of toxicity and profanity. As the results shown, taken $0.2$ as the threshold, our Qwen-VL is more harmfulless than PaLI-X.
>
> | RACE  |   toxicity < 0.2 |   0.2 <= toxicity <= 0.8 |   0.8 < toxicity |   profanity < 0.2 |   0.2 <= profanity <= 0.8 |   0.8 < profanity |
> |:----------------|-----------------:|:-------------------------:|:-----------------:|:------------------:|:--------------------------:|:------------------:|
> | Black |   84.1% | 15.9% | 0% |    99.7% |   0.3% |  0% |
> | East Asian |   86.2% | 13.8% | 0% |    99.2% |   0.8% |  0% |
> | Indian|   80.5% | 19.5% | 0% |    99.7% |   0.3% |  0% |
> | Latino_Hispanic |   86.3% | 13.7% | 0% |    99.3% |   0.7% |  0% |
> | Middle Eastern  |   86.4% | 13.6% | 0% |    99.3% |   0.7% |  0% |
> | Southeast Asian |   86.7% | 13.3% | 0% |    99.2% |   0.8% |  0% |
> | White |   84.9% | 15.1% | 0% |    99%   |   1%   |  0% |
>
> | AGE|   toxicity < 0.2 |   0.2 <= toxicity <= 0.8 |   0.8 < toxicity |   profanity < 0.2 |   0.2 <= profanity <= 0.8 |   0.8 < profanity |
> |:-------------|:-----------------:|:-------------------------:|:-----------------:|:------------------:|:--------------------------:|:------------------:|
> | 0-2|   68.8% | 31.2% | 0% |   100%   |   0%   |  0% |
> | 3-9|   83.4% | 16.6% | 0% |    99%   |   1%   |  0% |
> | 10-19   |   86.4% | 13.6% | 0% |    99.2% |   0.8% |  0% |
> | 20-29   |   85.9% | 14.1% | 0% |    99.4% |   0.6% |  0% |
> | 30-39   |   85.5% | 14.5% | 0% |    99.4% |   0.6% |  0% |
> | 40-49   |   85.7% | 14.3% | 0% |    99.2% |   0.8% |  0% |
> | 50-59   |   83.4% | 16.6% | 0% |    99.6% |   0.4% |  0% |
> | 60-69   |   86.6% | 13.4% | 0% |    99.7% |   0.3% |  0% |
> | >= 70 |   75.4% | 24.6% | 0% |    99.2% |   0.8% |  0% |
>
> | GENDER   |   toxicity < 0.2 |   0.2 <= toxicity <= 0.8 |   0.8 < toxicity |   profanity < 0.2 |   0.2 <= profanity <= 0.8 |   0.8 < profanity |
> |:-------------|:-----------------:|:-------------------------:|:-----------------:|:------------------:|:--------------------------:|:------------------:|
> | Male|   86%   | 14%   | 0% |    99.6% |   0.4% |  0% |
> | Female   |   83.8% | 16.2% | 0% |    99.1% |   0.9% |  0% |
>
> | Model | - | Threshold | Ethnicity (Low/High) (&darr;) | Age (Low/High) (&darr;) | Gender (Low/High) (&darr;) |
> |:-------------|:-----------------:|:-------------------------:|:-----------------:|:------------------:|:------------------:|
> | PaLI-X | Toxicity | $0.2$ | $35.8$%/$40.4$% | $33.9$%/$40.0$% | - |
> | Qwen-VL | Toxicity | $0.2$ | $\mathbf{13.3}$%/$\mathbf{19.5}$% | $\mathbf{13.4}$%/$\mathbf{31.2}$% | $14.0$%/$16.2$% |
> | PaLI-X | Profanity | $0.2$ | $5.1$%/$8.5$% | $3.5$%/$10.3$% | - |
> | Qwen-VL | Profanity | $0.2$ | $\mathbf{0.3}$%/$\mathbf{1.0}$% | $\mathbf{0.3}$%/$\mathbf{1.0}$% | $0.4$%/$0.9$% |
>
> ```text
> [1] Lees, Alyssa, et al. "A new generation of perspective api: Efficient multilingual character-level transformers." Proceedings of the 28th ACM SIGKDD Conference on Knowledge Discovery and Data Mining. 2022.
> [2] Chen, Xi, et al. "PaLI-X: On Scaling up a Multilingual Vision and Language Model." arXiv preprint arXiv:2305.18565 (2023).
> [3] Kärkkäinen, Kimmo, and Jungseock Joo. "Fairface: Face attribute dataset for balanced race, gender, and age." arXiv preprint arXiv:1908.04913 (2019).
> ```

---

> ### Author Response · Authors · 2023-11-21
> **Looking forward to your reply**
>
> Dear reviewer 2REz,
>
> We sincerely appreciate your efforts and valuable time spent reviewing our work, as well as your constructive contribution to improving the quality of our paper.
>
> Have our responses effectively addressed your concerns? If you still have any issues or new questions, please feel free to let us know so that we may continue the discussion.

---

> ### Author Response · Authors · 2023-11-22
> **Looking for discussion before the period ends**
>
> Dear Reviewer 2REz,
>
> This is a kind reminder about the approaching due date for the author-reviewer discussion period. We have provided responses to your questions above, including the clarification of some misunderstandings, restatement of our novelty and contributions, and a comprehensive study toward the fairness and bias of our model. We also made some improvements to our manuscript following your suggestions.  Given the limited time remaining, we would be very grateful if you could take the invaluable time to review our responses. If our responses have addressed your concerns, we would be most appreciative if you could consider changing your initial rating. If you still have any remaining concerns, we are also glad to continue this discussion with you during this final window.
>
> Best,
>
> Authors

---

### Official Review · Reviewer_8ZLo · 2023-11-02

**Soundness:** 3 good
**Presentation:** 4 excellent
**Contribution:** 3 good
**Rating:** 8
**Confidence:** 4

**Summary:**

This paper showcase the qwen-vl as a versatile LMM, being able to perceive and understand both texts and images. The  qwen-vl series contains a multitask finetuned 7B model and a chatbox trained with interleaved data. The modeling is similar to flamingo but the trainable parts are different in different stage. The model achieves reasonable generalist scores.

**Strengths:**

(1) Very clear pretraining data size and mixture weights that helps the general audience get a sense of the pretraining distribution, though the paper uses some internal data, which is understandable

(2) good ablation study over different parts, window attention for highres

(3) good experiment setups that consider sufficient academic benchmarks,

(4) well written and easy to follow

**Weaknesses:**

(1) seems missing generalist PaLI results. The PaLI-X authors also have multitask finetuned model for VQA and captioning mixtures separately.

(2) missing the design / motivation or ablation of which part being trained during different stage. The stage 2 unfreezes ViT is for adapting to higher solution?

**Questions:**

Please comment on the weakness

---

> ### Author Response · Authors · 2023-11-20
> **Response to Reviewer 8ZLo**
>
> **[Miss PaLI results]**
>
> Thanks for pointing this our. However, PaLI-X's multi-task finetuning remains treating different tasks (*e.g.*, image caption, VQA and etc.) seperately instead of as a whole (*i.e.*, one model to inference on all tasks). We are also glad to add more PaLI-X's multi-task finetuning results as specialist SOTAs for comparision.
>
> **[Motivation and ablation on the training part in different training stages]**
>
> First of all, we thank the reviewer for suggesting this ablation on the training part in different training stages. The motivation and intention of our choice on which part being tuned in different stage are listed as following:
>
> * For the first stage, where billion-scale image-text pairs are leveraged, we tune the vision part only due to: (i) Efficiency: Tuning solely the vision component is much more efficient than joint tuning both ViT and LLM. (ii) Potential for catastrophic forgetting: There exists a significant likelihood of the LLM suffering from catastrophic forgetting if we simultaneously fine-tune ViT and LLM without access to a dedicated pure-text corpus. Notebaly, compared to the text corpus used for LLM pre-training, text lengths in image-text pair dataset (*e.g.*, CC3M, CC12M, Laion, and etc.) are much more noisy and short. To overcome this catastrophic forgetting issue, in the second and third stages, we incorporate pure-text corpus when LLM is tuned.
> * For the second stage, the reasons behind jointly tuning ViT and LLM are two folds: (i) Additional Vision Tasks: We incorporate some additional vision tasks (*e.g.*, grounding and OCR) in this stage and all these tasks require fine-grained visual perception ability. To this end, we increase the input image resolution from $224$ to $448$ and tune ViT to adapt to this high resolution. (ii) Specific Output Requirements: Visual grounding asks the LLM to generate output in a specific behavior (*i.e.*, bounding box). Therefore, we choose to fune LLM to make this task doable.
> * In the third stage, the focus shifts primarily towards supervised fine-tuning for dialogue capacity, rather than introducing additional visual capacities (which have been incorporated into ViT during the previous two stages). Consequently, we opt to freeze the ViT component while keeping the remaining parts trainable.
>
> As per your request, we conduct additional ablation studies on the training part during the second stage. Given the limited amount of time and resources available to us in the response period, we are only able to get these experiments done in a small scale. Specifically, we employe a LLM with $1.8$B parameters and CLIP-ViT-L/$14$ as vision encoder, the overall parameters in our model are about $2$B. We train this model on the same dataset composition as we used in our paper, in spite of much fewer iterations. During the first stage, we tune the model with a batch size of $30720$ for $8000$ steps. During the second stage, in addition to the default setup, we also try to train our model with either ViT or LLM being frozen. All these three models for the second stage are tuned for $6000$ steps with a batch size of $2048$.
>
> | Benchmark | Qwen-VL($2$B) | Freeze ViT in Stage$2$ | Freeze LLM in Stage$2$ |
> |:---|:---:|:---:|:---:|
> | Nocaps (CIDEr) | $114.7$ | $109.7$ | $99.3$ |
> | VQAv2-val (VQA Score) | $74.6$ | $68.9$ | $64.4$ |
> | OKVQA-val (VQA Score) | $45.7$ | $43.2$ | $32.7$ |
> | VizWiz-val (VQA Score) | $29.4$ | $22.5$ | $19.3$ |
> | TextVQA-val (VQA Score) | $52.5$ | $27.7$ |  $40.1$ |
>
> There are several observations:
>
> * Compared to jointly tuning both ViT and LLM, freeze ViT or freeze LLM both lead to significant performance decrease.
> * On most benchmarks, freeze ViT shows better results than freeze LLM. This observation is in line with some current vision-language machines who prefer to tune LLM and keep the ViT frozen.
> * However, for text-oriented VQA(i.e, TextVQA), it's interesting to see that freeze LLM (tune ViT) shows significant higher accuracy compared to freeze ViT (tune LLM). We ascribe this phenomenon to text-oriented VQA requires more fine-grained visual perception capacity which has not been learnt in previous ViT training process(*e.g.*, vision-language contrastive learning in CLIP, and coarse-grained image captioning in the 1st stage).

---

> > ### Public Comment · ~Nick_Yang1 · 2023-11-27
> >
> > Will the 2B model be made available? I find it challenging to accept the results without this access. Additionally, the absence of the pre-training and multi-task pre-training code raises concerns about potential data leakage during the pre-training phase, which is not qualified for publication.

---

### Official Review · Reviewer_RE8P · 2023-11-06

**Soundness:** 3 good
**Presentation:** 2 fair
**Contribution:** 2 fair
**Rating:** 3
**Confidence:** 3

**Summary:**

This paper proposes QWEN-VL, a series of large-scale vision-and-language models (LVLMs). Further details can be found in Strengths.

**Strengths:**

- S1: This is one of a few open-source models where the model weights are released (though I don’t think the data is; there is also some “in-house” data; see Table 2). This can benefit the research community; the claim is that while the performance of QWEN-VL is still behind private models, it excels in the open-source community, especially in terms of capabilities it supports (Figures 4-7).

- S2: The training pipeline (Figure 3) is sound and simple.

**Weaknesses:**

- W1: Weak research significance and contributions. This work is a huge engineering effort and it is appreciated. However, research-wise, I am not convinced that it can provide any insights in terms of large-scale model training, architecture, or evaluation.

- W2: Weak discussion of related work and clarity: To make W1 worse, the paper does not properly discuss the relevant work. If the paper would like to focus on the open-source aspect, I think it can expand this part much more heavily. What are the existing open-source LVLMs and what are “open” about them? What are the capabilities they support and so on? However, based on the current presentation this is unclear.

**Questions:**

- Is the train-test overlap between benchmarks taken care of? Especially COCO-based datasets.

Please address the two points in my Weaknesses.

---

> ### Author Response · Authors · 2023-11-20
> **Response to Reviewer RE8P (1/2)**
>
> **[Research contributions and insights]**
>
> We thank the reviewer for valuable feedback. In addition to the contributions listed in our paper, we restate Qwen-VL's contribution, technical insights, and novelty point by point in the general response. We hope that our detailed reply will ease your concerns.
>
> **[Discussions of open-source work]**
>
> We thank your kind suggestion. In the below table, we compare Qwen-VL with previous open-source large vision-language models (LVLMs) in terms of their supporting language and capable task. As shown, Qwen-VL can support both English and Chinese and is capable to finish four kinds of vision-language tasks within the same model. Moreover, as mentioned in our paper, Qwen-VL also outperforms these predecessors across several benchmarks on these four tasks. Based on the versatile ability of Qwen-VL and its superior performance on a wide range of benchmarks, we believe that Qwen-VL can serve as a strong foundation for both future research and application purpose.
>
> *We trust our response can make the capabilities of previous LVLMs much more clear. We also include this discussion in our revised paper.*
>
> > Note: For text-oriented VQA, we refer to whether the model is designed or optimized to tackle this problem explicitly.
>
> | Model          | English | Chinese | Caption | General VQA | Text-oriented VQA | Grounding |
> |----------------|:---------:|:---------:|:---------:|:-----:|:------------:|:-----------:|
> | Kosmos         | $\checkmark$ |         | $\checkmark$ | $\checkmark$ |            |           |
> | BLIP2          | $\checkmark$ |         | $\checkmark$ | $\checkmark$ |            |           |
> | LLaVA          | $\checkmark$ |         | $\checkmark$ | $\checkmark$ |            |           |
> | MiniGPT-4      | $\checkmark$ |         | $\checkmark$ | $\checkmark$ |            |           |
> | ChatGLM        | $\checkmark$ | $\checkmark$ | $\checkmark$ | $\checkmark$ |            |           |
> | mPLUG-Owl      | $\checkmark$ |         | $\checkmark$ | $\checkmark$ |            |           |
> | InstructBLIP   | $\checkmark$ |         | $\checkmark$ | $\checkmark$ | $\checkmark$ |           |
> | mPLUG-DocOwl   | $\checkmark$ |         | $\checkmark$ | $\checkmark$ | $\checkmark$ |           |
> | Kosmos2        | $\checkmark$ |         | $\checkmark$ | $\checkmark$ |            | $\checkmark$ |
> | Shikra         | $\checkmark$ |         | $\checkmark$ | $\checkmark$ |            | $\checkmark$ |
> | Qwen-VL        | $\checkmark$ | $\checkmark$ | $\checkmark$ | $\checkmark$ | $\checkmark$ | $\checkmark$ |

---

> ### Author Response · Authors · 2023-11-20
> **Response to Reviewer RE8P (2/2)**
>
> **[Train-test overlap]**
>
> We appreciate the reviewer for highlighting the potential image contamination between our training dataset and various evaluation benchmarks, notably the coco-based benchmarks. To assess and address this concern, we conducted two contrast experiments. However, due to constraints in both time and resources within the response period, these experiments were performed on a smaller scale. Below, we outline the setups of our experiments:
>
> * We use a LLM with $1.8$B parameters and CLIP-ViT-L/$14$ as vision encoder. Total parameters are about $2$B.
> * We perform the first and second training stages, and evaluate the models on several caption/VQA/Grounding benchmarks.
> * For the first stage, we train the model with a batch size of $30720$ for $8000$ iterations. In other words, the total consumed image-text pairs are about $245.76$M.
> * For the second stage, we train our model with a batch size of $2048$ for $6000$ steps.
>
> For the baseline experiment, we use exact the same composition of datasets as in our paper. For the second experiment, we follow previous work to perform near-deduplication. Specifically, we deduplicate all coco images used in some coco-based evaluation benchmarks (*e.g.*, VQA, OKVQA, Refcoco/Refcoco+/Refcocog). The final results are shown in table below:
>
> | Benchmark | Is COCO-based? | Qwen-VL($2$B) | Qwen-VL($2$B) + Data Deduplication |
> |:---|:---:|:---:|:---:|
> | Nocaps (CIDEr) | | $114.7$ | $114.3$ |
> | VQAv2-val (VQA Score) | $\checkmark$ | $74.6$ | $74.3$ |
> | OKVQA-val (VQA Score) | $\checkmark$ | $45.7$ | $45.9$ |
> | VizWiz-val (VQA Score) | | $29.4$ | $29.8$ |
> | TextVQA-val (VQA Score) | | $52.5$ | $52.5$ |
> | RefCOCO-val (Accuracy) | $\checkmark$ | $79.9$ | $79.6$ |
> | RefCOCO+-val (Accuracy) | $\checkmark$ | $66.1$ | $66.5$ |
> | RefCOCOg-val (Accuracy) | $\checkmark$ | $74.2$ | $73.6$ |
>
> From the results, we observe little to no performance perturbation between two experiments for both coco-based and non coco-based benchmarks, which demonstrates the reliability of our results. We also appreciate the reviewer's professional comment.
>
> **[In-house data]**
>
> We thank the reviewer for approving our efforts in opening source our models. And we would like to further explain that despite some in-house data is indeed included in our training process (*e.g.*, Chinese image-caption dataset in Table $2$), we have made extensive efforts to provide detailed descriptions on construction and pretreating process of the datasets, the training mixture, and how the models are trained. We believe the information can be very helpful to experts who would like to reproduce our dataset construction as well as the whole training process at the same scale. Moreover, we are actively engaged in efforts to make additional information and data publicly available in the near future.

---

> ### Author Response · Authors · 2023-11-21
> **Do our responses address your concerns?**
>
> Dear reviewer RE8P,
>
> We appreciate your efforts and time in reviewing our work and participating in the rebuttal process to improve the quality of our paper.
>
> Have our responses adequately addressed your concerns? If you still have any issues with our responses or if there are any new questions, we are more than willing to continue the discussion with you.

---

> ### Author Response · Authors · 2023-11-22
> **Looking for discussion before the discussion period ends**
>
> Dear Reviewer RE8P,
>
> This is a kind reminder about the approaching due date for the author-reviewer discussion period. We have provided responses to your questions above and made some improvements to our manuscript following your suggestions. Given the limited time remaining, we would be very grateful if you could take the invaluable time to review our responses. If our responses have addressed your concerns, we would be most appreciative if you could consider changing your initial rating. If you still have any remaining concerns, we are also glad to continue this discussion with you during this final window.
>
> Best,
>
> Authors

---

### Author Response · Authors · 2023-11-20
**General Response: Contributions and New experiments (1/2)**

We sincerely appreciate the time and effort invested by all reviewers in evaluating our paper. We are pleased to note that the reviewers have acknowledged our contributions:

* Contributions to the research community [RE8P, 8ZLo, 2REz]
* A concise and innovative training pipeline [RE8P, 8ZLo]
* A comprehensive ablation study [8ZLo], along with satisfactory performance demonstrated across a wide range of benchmarks [8ZLo, 2REz]

We are also grateful to all reviewers for their insightful and constructive suggestions, which have greatly contributed to further improving our paper. In addition to addressing these comments point by point below, we would like to begin by restating our key contributions, technical insights, and the novelty presented in our work.

***[Contributions, technical insights, and novelty]***

* ***Open-source:*** Qwen-VL is an open-sourced large vision-language model that excels in **(i)** achieving leading performance across a wide range of vision-language understanding and generation tasks, **(ii)** offering multi-lingual support, particularly in English and Chinese, **(iii)** accommodating multi-image and high-resolution inputs, and **(iv)** demonstrating fine-grained visual perception abilities, particularly in scene text-oriented visual question-answering and visual grounding. We believe that our open-sourced models can serve as a robust research foundation for the fields of vision, vision-language, and embodied AI. For instance, researchers can leverage our model partially [1] or in its entirety [2] for further exploration in multi-modality learning.
* ***Practicable and referential training details:*** In our paper, we introduce a stage-wise training pipeline designed for efficient vision-language pre-training, multi-task fine-tuning, and supervised fine-tuning. Unlike previous representative vision-language models like PaLI-X, which leverages proprietary in-house data and utilize publicly inaccessible model weights (*e.g.*, ViT-22B), along with significantly high training costs, our Qwen-VL's training process is more practical and holds considerable referential significance for future research. Furthermore, although we incorporate a small portion of in-house data, we have made extensive efforts to provide comprehensive details regarding the training process, data preprocessing, and data construction. We believe that this detailed information will be valuable for experts who want to replicate our entire training procedure.
* ***Higher resolution inputs:*** To the best of our knowledge, Qwen-VL stands out as one of the few open-source initiatives that enhance the competency of multimodal chatbots by harnessing the potential of higher resolution images. While the PaLI series also incorporates high-resolution images to develop a more generalized agent, previous open-source vision-language models have either overlooked input resolution or employed specific techniques only for certain tasks that demand higher resolution inputs. There are also several works [3, 4] which have delved deeper into the utilization of high resolution in LVLMs. We believe Qwen-VL can serve as a strong basement toward this direction.
* ***Focusing on conventional benchmarks and instruction-following evaluation in the same time:*** In our paper, we conduct a comprehensive evaluation of Qwen-VL across various vision-language tasks. We select several diverse conventional benchmarks to assess our model from different perspectives, including tasks such as image captioning, general visual question answering (VQA), text-oriented VQA, and visual grounding. Additionally, we perform evaluations on recently proposed instruction-following benchmarks to gauge our model's performance in real-world applications. To the best of our knowledge, our research stands out as one of the few focusing on evaluating models across both conventional tasks and instruction-following capacities in the realm of vision-language understanding.

```text
[1] Yao, Zhewei, et al. "DeepSpeed-VisualChat: Multi-Round Multi-Image Interleave Chat via Multi-Modal Causal Attention." arXiv preprint arXiv:2309.14327 (2023).
[2] Chen, Xinyu, et al. "QwenGrasp: A Usage of Large Vision Language Model for Target-oriented Grasping." arXiv preprint arXiv:2309.16426 (2023).
[3] Li, Bo, et al. "OtterHD: A High-Resolution Multi-modality Model." arXiv preprint arXiv:2311.04219 (2023).
[4] Bavishi, Rohan, et al. "Introducing our multimodal models" 2023.
```

---

> ### Author Response · Authors · 2023-11-20
> **General Response: Contributions and New experiments (2/2)**
>
> ***[Additional experiments]***
>
> We also summarize supporting experiments in our response according to reviewers' suggestions:
>
> * Ablation studies on image overlap between train datasets and benchmarks, especially the coco-based benchmarks. [RE8P]
> * Ablation studies on the training part of Qwen-VL in the second training stage. [8ZLo]
> * Analysises on the model's fairness and bias, and comparison with other methods. [2REz]
>
> We have included all additional experiments and modifications in our revision, which can be found in the appendix. We hope our responses provided below could clarify all reviewers' confusion and alleviate all concerns. We sincerely thank the reviewers for their valuable time once again.
>
> Best,
>
> Authors

---

> ### Public Comment · ~Nick_Yang1 · 2023-11-27
>
> Regarding open-sourcing, I found no available code for reproducing the results, particularly for the training process. The paper mentions that the model was trained using model parallelism, yet there's no supporting evidence or code provided to corroborate this claim. Moreover, the concept of using higher resolution seems overly simplistic and has been commonly employed in previous vision-language or purely vision tasks. Could the authors perhaps provide the complete code to train the model from scratch for full transparency and reproducibility?

---

### Meta-Review · Area_Chair_uCUE · 2023-12-14

**Metareview:**

Despite the effort in open-sourcing the model and its weights, the reviewers find QWEN-VL lacking in significant research contributions and technical novelty. While some aspects like the training pipeline and ablation studies are appreciated, the paper falls short in justifying its claims and demonstrating advancements in the field.

The paper fails to provide concrete insights into large-scale model training, architecture, or evaluation. Its focus on engineering effort without substantial theoretical or methodological contributions weakens its academic value. It also lacks a comprehensive discussion of existing open-source LVLMs and their capabilities, hindering comparisons and highlighting the unique value proposition of QWEN-VL.

Reading the paper feels like a technical report: The motivation for specific training stages and the ablation study design are not adequately explained, raising questions about the model's inner workings and the effectiveness of its components.

**Justification For Why Not Higher Score:**

I recommend the authors strengthen the research significance by providing insights into model training, architecture, or evaluation. Expand the discussion of related work, especially on open-source LVLMs and their capabilities.

Most importantly, as a research publication, please clearly explain the design rationale and provide more detailed explanations of the ablation study.

**Justification For Why Not Lower Score:**

N/A

---

### Decision · Program_Chairs · 2024-01-16

Reject